# Exploring the Distinct Binding and Activation Mechanisms for Different CagA Oncoproteins and SHP2 by Molecular Dynamics Simulations

**DOI:** 10.3390/molecules26040837

**Published:** 2021-02-05

**Authors:** Quan Wang, Wen-Cheng Zhao, Xue-Qi Fu, Qing-Chuan Zheng

**Affiliations:** 1Edmond H. Fischer Signal Transduction Laboratory, College of Life Sciences, Jilin University, Changchun 130023, China; wangquan18@mails.jlu.edu.cn (Q.W.); wczhao16@mails.jlu.edu.cn (W.-C.Z.); 2Laboratory of Theoretical and Computational Chemistry, Institute of Theoretical Chemistry, International Joint Research Laboratory of Nano-Micro Architecture Chemistry, Jilin University, Changchun 130023, China

**Keywords:** SHP2, EPIYA, *H. pylori*, MD simulations, allosteric

## Abstract

CagA is a major virulence factor of *Helicobacter pylori*. *H. pylori* CagA is geographically subclassified into East Asian CagA and Western CagA, which are characterized by the presence of a EPIYA-D or EPIYA-C segment. The East Asian CagA is more closely associated with gastric cancer than the Western CagA. In this study, molecular dynamic (MD) simulations were performed to investigate the binding details of SHP2 and EPIYA segments, and to explore the allosteric regulation mechanism of SHP2. Our results show that the EPIYA-D has a stronger binding affinity to the N-SH2 domain of SHP2 than EPIYA-C. In addition, a single EPIYA-D binding to N-SH2 domain of SHP2 can cause a deflection of the key helix B, and the deflected helix B could squeeze the N-SH2 and PTP domains to break the autoinhibition pocket of SHP2. However, a single EPIYA-C binding to the N-SH2 domain of SHP2 cannot break the autoinhibition of SHP2 because the secondary structure of the key helix B is destroyed. However, the tandem EPIYA-C not only increases its binding affinity to SHP2, but also does not significantly break the secondary structure of the key helix B. Our study can help us better understand the mechanism of gastric cancer caused by *Helicobacter pylori* infection.

## 1. Introduction

The gastric bacterium *Helicobacter pylori* was discovered for the first time in 1984 and has attracted more and more attention in recent decades [1]. Previous studies have shown that chronic infection with *H. pylori* is strongly associated with various diseases, including gastric inflammation, peptic ulcers and gastric cancer [2]. In particular, the incidence of gastric cancer in East Asian countries is eight-fold greater than that in North America, which is strongly related to the infection with *H. pylori* [3]. The cytotoxin-associated gene A (CagA) is a major factor of virulence of *H. pylori*, which plays a very important role in the pathogenic process [4]. The *cagA* gene is located at one end of a 40-kb genomic DNA segment of *H. pylori*, which is also known as the *cag* pathogenicity island (*cag* PAI) [5], and in this region several genes encode proteins composing a type IV secretion system (TFSS), which forms a syringe-like structure that is capable of delivering CagA across bacterial membranes [6]. Upon delivery into host cells, CagA is tethered to the inner leaflet of the plasma membrane. Then, membrane-localized CagA undergoes tyrosine phosphorylation at the C-terminal Glu-Pro-Ile-Tyr-Ala (EPIYA) motif by Src family kinases (SFK) such as c-Src, Yes, Fyn, and Lyn [7,8], and some other kinases such as c-Abl [9]. Based on the difference in the sequences flanking the EPIYA motif, four distinct EPIYA segments have been identified, namely EPIYA-A, -B, -C and -D. Each of them contains a single EPIYA motif [10]. The C-terminal EPIYA-repeat region of Western CagA is in an arrangement of EPIYA-A, EPIYA-B and a variable number (mostly 1–3) of EPIYA-C segments in tandem (termed ABC-type CagA) (Figure 1A). The C-terminal EPIYA-repeat region of East Asian CagA is in an arrangement of EPIYA-A, EPIYA-B and EPIYA-D segments in tandem (termed ABD-type CagA) (Figure 1A) [11].

CagA can perform multiple functions in tyrosine phosphorylation-dependent and -independent manners in the host cells. In tyrosine phosphorylation-independent manners, the N-terminal region of CagA is capable of interacting with several proteins, such as the gastric tumor suppressor RUNX3, which may additionally contribute to oncogenesis [12]. The tyrosine-phosphorylated EPIYA segments can serve as docking sites for a series of proteins, especially those possessing the Src homology 2 (SH2) domains [13]. In tyrosine phosphorylation-dependent manners, the EPIYA-B segment is able to bind to the p85 subunit of phosphatidylinositol 3-kinase (PI3K) and deregulates the PI3K/AKT pathway [14]. Remarkably, CagA specifically interacts with the Src homology 2 (SH2) domain-containing protein tyrosine phosphatase 2 (SHP2) in a tyrosine phosphorylation-dependent manner [15]. The formation of the CagA–SHP2 complex is mediated by the interaction between the tyrosine-phosphorylated EPIYA-C or EPIYA-D segment and the SH2 domains of SHP2 [16].

SHP2 is a ubiquitous protein tyrosine phosphatase with conserved structure and function, and participates in many biological processes [17,18]. The dysregulation of SHP2 has been implicated in the pathogenesis of many human diseases, such as Noonan syndrome (NS), LEOPARD syndrome (LS), and multiple types of cancers, including leukemia, lung and breast cancer, liver cancer and neuroblastoma [17,19,20,21]. SHP2 is composed of two tandem SH2 domains at the N-terminal (N-SH2 and C-SH2), a central PTP domain, and a disordered C-terminal tail (Figure 1B) [22]. In the resting state, the N-SH2 and the PTP domains of SHP2 can form inhibitory intramolecular interactions and keep SHP2 in a closed, inactive state. The tyrosine phosphorylated upstream activators binding to the SH2 domains induce the conformation rearrangement and disrupt the autoinhibitory interface; SHP2 is then in an open, active conformation and can dephosphorylate its substrates [23,24,25]. Both tyrosine-phosphorylated EPIYA-C and EPIYA-D segments may prefer to binding to the N-SH2 domain of SHP2, but the effects of binding are quite different. A single EPIYA-C segment could only weakly activate SHP2, while a single EPIYA-D segment was enough to activate SHP2 strongly. The EPIYA-D segment could bind to the N-SH2 domain of SHP2 two orders of magnitude greater than EPIYA-C, and change the conformation of SHP2 to an open state. In other words, East Asian CagA can bind to SHP2 and strongly activate SHP2 via a high-affinity monovalent interaction with N-SH2. Interestingly, duplications of EPIYA-C in Western CagA can enable divalent high-affinity binding with SHP2 via the N-SH2 and C-SH2 domains [26]. These suggest distinct mechanisms for CagA and SHP2 deregulation. It is important for us to give an insight into the details of the regulation mechanisms for CagA and SHP2.

In this study, molecular dynamics (MD) simulations and molecular mechanics–generalized Born surface area (MM-GB/SA), which have been proven to be powerful and valuable tools, were performed to explore the binding details of SHP2 and EPIYA. In this article, we mainly focus on three points: (1) How the phosphorylated EPIYA-C and EPIYA-D combine with SHP2. (2) Why a single EPIYA-D peptide can activate SHP2, but a single EPIYA-C peptide cannot. (3) Explore the allosteric regulation mechanism between SHP2 and EPIYA. Our work may provide useful information to explain the allosteric mechanism of SHP2, and give a new method of anti-cancer therapy.

## 2. Results and Discussion

### 2.1. The Molecular Docking Analysis

First, ZDOCK was used to obtain the initial docking poses, and then RDOCK was used to optimize and choose the final docking poses (Table 1). The structure inspection revealed there is no unreasonable structure. Compared with the crystal structures (PDB codes: 5 × 7B, 5 × 94), the RMSD values of the RDOCK results are all less than 1.5Å. These results indicate that the docking results are credible, and can be used as the initial structure for subsequent MD simulations.

### 2.2. The Overall Structural Properties

In order to investigate the overall structural stability, root-mean-square deviations (RMSD) and root-mean-square fluctuations (RMSF) of the backbone atoms for all systems were calculated referring to the initial structure (Figure 2).

As shown in Figure 2A, the RMSD values of all systems are pretty stable during the whole simulation, which indicates that there are no obvious structure changes having occurred under the simulations in all systems. In addition, in order to obtain a better understanding of the effect of phosphorylated EPIYA on different domains of SHP2, the RMSD values of the N-SH2 domain (residues 6 to102), the C-SH2 domain (residues 110 to 216), and the PTP domain (residues 247 to 517) were calculated separately for all systems. As shown in Figure 2B, The RMSD values and fluctuations of the PTP domain are more stable than those of the N-SH2 and C-SH2 domains, which means that the PTP domain could be very stable under the simulations in all systems. This result shows that the binding of phosphorylated EPIYA (no matter whether EPIYA-C or EPIYA-D) has no direct effect on the overall structure of the PTP domain. To investigate the changing flexibility of SHP2, the RMSF values of all systems were calculated (Figure 2A). On the whole, the C-SH2 domain is more flexible than the N-SH2 domain, and some loops in the PTP domain also show a high level of flexibility. Although the overall trends of RMSF curves are similar, the binding of phosphorylated EPIYA still has effects on some key regions of SHP2. Taking system EPIYA-D-N as an example, the RMSF values of the C-SH2 domain are higher than other systems. The CD loop and WPD loop also have more flexibility than other systems. However, the flexibility of the E loop and pY loop is weaker than that of the EPIYA-C-N system.

As shown in the results of RMSD and RMSF, the flexibility of the C-SH2 domain is much stronger than that of the N-SH2 and PTP domains. The binding of phosphorylated EPIYA could have an impact on the overall structure of SHP2, rather than just being limited to its binding site (or binding domain). These results may provide some help to explain the allosteric mechanism of SHP2.

### 2.3. MM-GB/SA Calculations and Energy Decomposition Analysis

To further explore the interaction between SHP2 and EPIYA, MM-GB/SA calculations and energy decomposition were performed for all complex systems. As shown in Table 2, the binding affinity of EPIYA-D is much stronger than EPIYA-C, whether it binds to the N-SH2 or C-SH2 domains. For peptide EPIYA-D, its binding affinity to the N-SH2 domain is the strongest in all complex systems, and its binding affinity to the C-SH2 domain is slightly weaker than that of the N-SH2 domain (still stronger than other systems). For peptide EPIYA-C, its binding free energy values for the N-SH2 and C-SH2 domains are −17.06 kcal mol^−1^ and −19.87 kcal mol^−1^, respectively, which are much weaker than EPIYA-D. This decrease in binding affinity is mainly caused by the enthalpy, but the entropy also plays an important role in it. For system EPIYA-C-DUAL, the binding free energy values of the N-SH2 and C-SH2 domains were calculated separately (Table 2). Interestingly, when two EPIYA-C simultaneously bind to SHP2, the binding affinity of the EPIYA-C and N-SH2 domains is greatly enhanced. This increase in affinity is mainly due to the decrease in entropy. This change in binding affinity is probably the reason why the tandem EPIYA-C peptides could activate SHP2.

Furthermore, to understand the binding mechanism of EPIYA, a per residue decomposition analysis of individual amino acid residues was performed. Residues with a contribution over −2 kcal mol^−1^ will be discussed emphatically. For the EPIYA-D-N system (Appendix A), the electronegative PTR is firmly surrounded by the electropositive residues ARG32, LYS35 and LYS55. In addition, SER34, SER36 and THR42 also assist these electropositive residues in forming the binding pocket for PTR (Figure 3A). Residue D4 could form an electrostatic interaction with LYS89 and LYS91. In addition, residues I3 and F5 also make a contribution to the binding of EPIYA-D because of their strong hydrophobicity. For the system EPIYA-C-N (Figure 3B, Appendix A), it has the same binding pocket for PTR as the system EPIYA-D-N. However, the contribution of TYR66 and GLY68 disappears because D5 instead of F5 exists at the pY + 5 position. For the system EPIYA-D-C (Figure 3C, Appendix A), the PTR binding pocket is formed by residues ARG138, GLN141, SER140, VAL148 and HIS169. Residues I3 and F5 also promote binding by their hydrophobic interaction. Interestingly, there is an electrostatic interaction between D6 and ARG186. For the system EPIYA-C-C (Figure 3D, Appendix A), it almost has the same binding mode as system EPIYA-D-C, but the contribution of residue ARG186 disappears because I6 instead of D6 exists at the pY + 6 position. For the system EPIYA-C-DUAL (Appendix A), the binding mode of the tandem EPIYA-C is the same as that of the single EPIYA-C, but the interaction between SHP2 and EPIYA-C would be closer in the system EPIYA-C-DUAL.

Interestingly, whether it is EPIYA-C or EPIYA-D, their binding is mainly dependent on the PTR and subsequent residues (PTR0 to pY + 6). The residues (−6 to −1) located ahead of the PTR on the peptides make almost no contribution to the binding. This part of the EPIYA peptides should have no effect on the activation of SHP2, and swing freely under the simulations.

### 2.4. Hydrogen Bond Network Analysis

Hydrogen bond analysis is an important method for analyzing the interaction between (or inside) proteins. Here, the hydrogen bonds between SHP2 and EPIYA are calculated to obtain the binding details, and the hydrogen bonds inside SHP2 are calculated to monitor the internal structural changes of SHP2.

There are some constant hydrogen bonds between SHP2 and EPIYA. For the N-SH2 domain (Figure 4A,C and Appendix A), PTR could form hydrogen bonds with residues ARG32, SER34, LYS35, SER36 and THR42. Residue A1 could form a hydrogen bond with HIS53, T2 could form a hydrogen bond with LYS91, and D4 could form a hydrogen bond with LYS89, but compared with EPIYA-C, EPIYA-D has more hydrogen bond interactions with the N-SH2 domain. For the C-SH2 domain (Figure 4B,D and Appendix A), PTR could form hydrogen bonds with residues ARG138, SER140, GLN141 and SER142. Residue A1 could form a hydrogen bond with HIS169, and T2 could form a hydrogen bond with THR205. EPIYA-D still has more hydrogen bond interactions with the C-SH2 domain.

The hydrogen bonds inside SHP2 were calculated to monitor the conformational changes of SHP2. Here, the hydrogen bonds between the three domains and the key helix B are discussed. The change in hydrogen bonds between the N-SH2 and PTP domains could reflect the structural changes in the autoinhibition site. In system SHP2 (Figure 5A, Appendix A), ASN58 forms a hydrogen bond with GLN506, GLY60 forms a hydrogen bond with GLN510, and ALA72 forms a hydrogen bond with GLN506. THR73 forms a hydrogen bond with GLU258, GLU76 forms a hydrogen bond with ARG265, and ASP61 forms a hydrogen bond with ARG465. For system EPIYA-D-N (Figure 5B, Appendix A), the hydrogen bonds THR73-GLU258, GLU76-ARG265 and ASP61-GLY465 disappear, but the residue ASP61 forms new hydrogen bonds with ALA461 and GLY464. As shown in Figure 5E, when EPIYA-D binds to the N-SH2 domain of SHP2, the key residue ASP61 undergoes a 90-degree flip. The system EPIYA-C-N (Figure 5C, Appendix A) has similar hydrogen bond interactions between the N-SH2 and PTP domains. In this system, ASP61 does not flip, but GLU76 forms a hydrogen bond with SER502 instead of ARG265. The system EPIYA-C-DUAL (Figure 5D, Appendix A) has a similar hydrogen bond network to system EPIYA-D-N, but ASP61 could form new hydrogen bonds with resides GLY462 and ILE463. In addition, the residue ASP61 also undergoes a flip in this system (Figure 5E).

Helix B is the key helix which connects three domains, and it is also the important target for the allosteric drug study [27,28]. Compared with the system SHP2 (Figure 6A and Appendix A), EPIYA-D binding to the N-SH2 domain of SHP2 could cause the rearrangement of the hydrogen bonds of helix B (Figure 6B and Appendix A). The hydrogen bond LYS35-GLU249 disappears because LYS35 is firmly fixed by PTR and keeps it away from the helix B. The hydrogen bond GLU258-THR73 disappears, but the new hydrogen bond GLU258-ARG498 is formed. The hydrogen bond interaction between ARG4 and GLU252 becomes weaker, but ARG4 forms new hydrogen bonds with GLN256 and GLU258. These hydrogen bond changes are mainly due to the deflection of the key helix B (Appendix A). For system EPIYA-C-N (Figure 6C, Appendix A), the hydrogen bonds of helix B are also different from system SHP2, but this difference is due to the disappearance of part of the secondary structure of helix B (Appendix A). For system EPIYA-C-DUAL (Figure 6D, Appendix A), its hydrogen bond interactions with helix B are similar to those of system EPIYA-D-N.

The binding of peptides EPIYA (EPIYA-D or EPIYA-C) could affect the interaction among three domains inside SHP2. For systems EPIYA-D-N and EPIYA-C-DUAL, helix B could move to squeeze the N-SH2 and PTP domains to destroy the autoinhibition pocket. However, for system EPIYA-C-N, its helix B cannot play the same role because its secondary structure is destroyed (Appendix A). This structural change may explain why the single EPIYA-C cannot activate SHP2.

### 2.5. Principal Component Analysis and Binding Free Energy Landscape

The PCA can give an insight into the conformational difference between different systems through the correlated motions of amino acid residues. Here, the binding free energy landscape was calculated by PC1 and PC2 as the two order parameters. As shown in Figure 7A, the system SHP2 has only one basin, which indicates that protein SHP2 only has one motion mode under the simulation. For system EPIYA-D-N, EPIYA-D binding to N-SH2 completely reconstructs its FEL (Figure 7B). There are five basins in this FEL result, which indicates that it has more motion modes than system SHP2 under the simulation. There are three basins in system EPIYA-C-N (Figure 7C) and five basins in system EPIYA-C-DUAL (Figure 7D). However, these five basins are not very distinct compared with system EPIYA-D-N. The EPIYA peptide’s binding to SHP2 could change the motion modes of SHP2. EPIYA-D binding to the N-SH2 domain of SHP2 could give SHP2 more motion modes, so that SHP2 can undergo more conformational changes. The tandem binding of EPIYA-C to the N-SH2 and C-SH2 domains simultaneously could give rise to the same result, although this result is not obvious.

### 2.6. Porcupine Plots Analysis

The PCA analysis results can be well understood from the porcupine plots figures. The highest Eigenvalue (PC1) of each system was chosen to draw the porcupine plots figures. The relative movement is reflected by the length and direction of the arrows. It should be noted that, although these motion modes are not the actual movement of SHP2, they are the motion modes that have the greatest impact on SHP2′s movement. The porcupine plots figure of system SHP2 will be used as a reference (Figure 8A). As shown in Figure 8, on the whole, the residues on the N-SH2 or C-SH2 domains tend to move in units of domains, and the motion mode of the PTP domain is more complicated. Compared with system SHP2 (Figure 8A), peptide EPIYA-D binding to the N-SH2 domain of SHP2 completely changes the motion modes of SHP2. As shown in Figure 8B, the motion modes of the N-SH2 domain change from anti-clockwise to clockwise, while the motion mode of the C-SH2 domain changes from clockwise to anti-clockwise. The motion mode of the PTP domain also changes. In addition, the motion mode of the autoinhibition pocket also tends to break the autoinhibition of SHP2. For system EPIYA-C-N (Figure 8C), the motion mode of the N-SH2 domain becomes disordered, but the motion modes of the C-SH2 and PTP domains are the same as that of system SHP2. System EPIYA-C-DUAL (Figure 8D) has a similar motion mode to system EPIYA-D-N, but its motion mode is more disordered, although it could also reduce the stability of the autoinhibition pocket.

Porcupine plots analysis can give an intuitive view of the effect of peptide EPIYA on the motion modes of SHP2. The changes in the motion mode of the autoinhibition pocket can help us understand the allosteric regulation mechanism of SHP2. In addition, these results may explain why a single peptide EPIYA-D can activate SHP2, but a single peptide EPIYA-C cannot.

### 2.7. Correlational Analysis

In order to better understand the effect of EPIYA on the conformational changes of SHP2, we calculated the dynamic cross-correlation maps of the C_α_–C_α_ displacement. The red and yellow parts are called the “positive region”, and they represent these pairs of residues moving in the same direction. The blue part is called the “negative region”, and it represents these pairs of residues moving in the opposite direction. EPIYA-D binding to the N-SH2 domain of SHP2 significantly changes the motion correlation of SHP2 (Figure 9A,B). The negative regions of the N-SH2 domain decrease, and the negative regions of the C-SH2 domain increase. At the same time, the negative regions of the PTP domain decrease, and the positive regions increase. EPIYA-C binding to the N-SH2 domain of SHP2 has little effect on the motion correlation of SHP2, and just slightly increases the negative regions of SHP2 (Figure 9C). The system EPIYA-C-DUAL (D) has a similar correlation map to system EPIYA-C-N, except for the fact that there are fewer positive regions.

## 3. Materials and Methods

### 3.1. Preparation of the Structures

The crystal structures of protein SHP2 (residues 1 to 526, PDB code 4DGP [29]) and phosphorylated peptides EPIYA-C (sequence VSPEPIpYATIDDL, PDB code: 5X7B [26]) and EPIYA-D (sequence ASPEPIpYATIDFD, PDB code: 5X94 [26]) were taken from the Protein Data Bank. Unless otherwise specified, EPIYA-C and EPIYA-D always indicate the phosphorylated form of the motifs. The missing residues were modeled by software MODELLER [30]. In this study, protein–peptides docking was used to construct the different complex systems with the software Discovery Studio [31]. The SHP2 crystal structure 4DGP [29] was chosen as the initial structure, and the EF loops of the SH2 domains were adjusted to accommodate the phosphopeptide according to the crystal structures of the SH2 domains and phosphorylated EPIYA [26]. The details of the docking site were obtained from the co-crystallization of SH2 domains and phosphorylated EPIYA (PDB code: 5X7B, 5X94) [26]. ZDOCK [32] was used to obtain the initial docking poses, and RDOCK [33] was used to optimize and screen out the final docking poses. In this study, six systems were prepared: system SHP2 (SHP2 without ligand), system EPIYA-D-N (EPIYA-D binding to the N-SH2 domain of SHP2), system EPIYA-D-C (EPIYA-D binding to the C-SH2 domain of SHP2), system EPIYA-C-N (EPIYA-C binding to the N-SH2 domain of SHP2), system EPIYA-C-C (EPIYA-C binding to C-SH2 domain of SHP2), and system EPIYA-DUAL (one EPIYA-C binding to N-SH2 domain and another EPIYA-C binding to C-SH2 domain). In order to better distinguish the residues in proteins and peptides, the residues in SHP2 will be identified by three letters, and the residues in EPIYA will be identified by one letter (except the phosphotyrosine, PTR). In addition, the number of residues of EPIYA peptides has been reconsidered, and the number of phosphotyrosines is designated as the number zero. The protonation states of all systems were assigned based on the calculation results of the H++ online website [34].

### 3.2. Molecular Dynamics Simulations

MD simulations were carried out using the AMBER 16 software package [35] and ff14SB force field [36]. The force field of phosphotyrosine was obtained from the AMBER parameter database [37,38]. In order to keep the whole systems in an electric neutral state, sodium ions (Na^+^) and chloride ions (Cl^−^) were added using the t-LEaP module [35]. All systems were solvated with TIP3P water model in a truncated octahedron with a 10Å cutoff between the complexes and a box boundary under the simulations [39]. The complex structures were initially fixed with a 500 kcal mol^−1^ Å^−2^ constraint, and the solvent and ions were submitted to 12,000 steps of steepest decent (SD) minimization followed by 10,000 steps of conjugate gradient (CG) minimization for all systems. Subsequently, the minimization was repeated for 12,000 steps of SD and 10,000 steps of CG without restraints. Thereafter, the systems were gradually heated from 0 to 310 K, applying harmonic restraints with a force constant of 10.0 mol^−1^ Å^−2^ on the solute atoms, and then the equilibration was performed for 5000 ps. Finally, a total of 1000 ns MD simulation was simulated for each system under NPT ensemble conditions using periodic boundary conditions and particle mesh Ewald [40] for long range electrostatics. The temperature was maintained at 310K by coupling to a Langevin heat bath using a collision frequency of 1 ps^−1^, and a constant isotropic pressure was maintained at 1 bar using the Berendsen barostat [41]. All bonds involving hydrogen atoms were held fixed using the SHAKE algorithm [42]. In this study, all visualization of the structures and trajectories was done by software packages PyMOL [43], VMD [44], and Chimera [45]. Most of the MD data analyses were performed by the *cpptraj* module of AMBER 16. The binding free energy was calculated using the molecular mechanics–generalized Born surface area (MM-GB/SA) method by AMBER 16. The porcupine plots analysis was performed by the ProDy interface in the VMD software.

### 3.3. Analysis of Hydrogen Bonds

Hydrogen bonds in all systems were analyzed using the *cpptraj* module of AMBER16. The presence of these bonds was calculated over the last 300 ns of the simulations. The criteria used for hydrogen bonding were that the distance was less than 3.5 Å between the hydrogen bond acceptor and the hydrogen bond donor, and that the angle of the acceptor-H-donor was more than 120°.

### 3.4. Free Energy Calculations

The molecular mechanics–generalized Born surface area (MM-GB/SA) method [46,47,48] implemented in AMBER 16 [35] software was performed to calculate the binding free energy of the complex systems. The calculation formulas are as follows:(1)$$G_{\mathrm{bind}}=G_{\mathrm{complex}}-\left(G_{\mathrm{receptor}}+G_{\mathrm{ligand}}\right)$$
(2)$$G_{\mathrm{bind}}=E_{\mathrm{MM}}+G_{\mathrm{sol}}-\mathrm{TS}$$
(3)$$E_{\mathrm{MM}}=E_{\mathrm{ele}}+E_{\mathrm{vdW}}+E_{\mathrm{int}}$$
(4)$$G_{\mathrm{sol}}=G_{\mathrm{PB}/\mathrm{GB}}+G_{\mathrm{SA}}$$

In Equation (2), the E_MM_, G_sol_, and TS represent the molecular mechanics component in the gas phase, the stabilization energy due to solvation, and the vibrational energy term. In Equation (3), E_MM_ is the summation of E_int_, E_ele_, and E_vdW_, which are the internal interaction term, the coulomb interaction term, and the van der Waals interaction term, respectively. G_sol_ is the solvation contribution, and it can be separated into polar solvation energy (G_GB_) and nonpolar solvation energy (G_SA_). The G_GB_ can be calculated by the generalized Boltzmann method [49]. G_SA_ could be calculated by
(5)$$G_{\mathrm{SA}}=\gamma \mathrm{SASA}+\beta$$

In Equation (5), the γ and β, two empirical constants, were set as 0.0072 kcal mol^−1^ Å^−2^ and 0.00 kcal mol^−1^, respectively. The SASA is the solvent accessible surface area determined by a probe radius of 1.4 Å. The binding free energies were the average values of calculating 5000 snapshots sampling from the last 100 ns of the trajectories for the complex systems. The entropy is generally calculated using normal-mode analysis [50] by AMBER 16 software package. Since the normal-mode analysis is computationally expensive, only 100 snapshots from the 5000 snapshots were chosen to calculate the entropy.

### 3.5. Principal Component Analysis and Free Energy Landscape

Principal component analysis (PCA) [51,52] is a widely used method to understand the dynamics of biological systems, and especially to study the allosteric regulation mechanism. The PCA calculations on MD trajectories were performed starting from the first frame of the production run, and this analysis was done on all non-hydrogen atoms of SHP2. To generate the PCA data for all systems, the ions and water molecules were stripped from the MD trajectories by *CPPTRAJ* [35], and each frame of the MD simulation was superimposed onto the initial structure before the calculation of the covariance matrix. The covariance matrix *C* was calculated from the superimposed Cartesian coordinates of the ensemble of protein structures.
(6)$$c_{ij}=<\left(r_{i}-<r_{i}>\right)>\cdot <\left(r_{i}-<r_{i}>\right)>\left(i,j=1,2,3,\ldots ,3N\right)$$

In Equation (6), $r_{i}$ is a cartesian coordinate of the ith atom, $<r_{i}>$ represents the average of the atomic positions, and *N* is the number of the selected atoms. The top two principal components PC1 and PC2 were projected on the 3D coordinates of protein. Furthermore, the porcupine plots were generated using the ProDy [53] interface in the VMD [44] software.

Free energy landscape (FEL) is a useful method to study the allosteric regulation in proteins, and could show the energy change during the conformational transition process of the SHP2 [54,55,56]. In the FEL, free energy minima represent stable conformations while the energy barriers connecting the minima represent metastable states. The Gibbs free energy ($G_{i}$) is defined as follows:(7)$$G_{i}=-k_{B}T\ln(\frac{N_{i}}{N_{\max}})$$

In this Equation, $k_{B}$ is the Boltzmann’s constant, *T* is the absolute temperature, $N_{i}$ is the probability density of the MD trajectories, and $N_{\max}$ is the maximum probability for a state.

## 4. Conclusions

Chronic infection with *H. pylori* is strongly associated with gastric cancer. In particular, the incidence of gastric cancer in East Asian countries is eight-fold greater than that in North America, which is strongly related to the infection with *H. pylori*. This situation is mainly caused by the difference between EPIYA-D and EPIYA-C. In this study, 1000 ns molecular dynamics (MD) simulations and molecular mechanics–generalized Born surface area (MM-GB/SA) free energy calculation were performed to explore the allosteric regulation mechanism of SHP2. EPIYA-D possesses the best binding affinity to the N-SH2 domain of SHP2, and the second best binding affinity to the C-SH2 domain. The binding affinity of a single peptide EPIYA-C is very poor. However, the tandem binding of EPIYA-C simultaneously to the N-SH2 and C-SH2 domains can greatly enhance the binding affinity of the EPIYA-C and N-SH2 domains. This may explain why the tandem EPIYA-C can activate SHP2, but a single EPIYA-C cannot. The structural analysis shows that EPIYA-D binding to the N-SH2 domain could induce conformational changes in SHP2. The key helix B squeezes the N-SH2 and PTP domains to reduce the stability of the autoinhibition pocket. When a single EPIYA-C binds to the N-SH2 domain, the secondary structure of the key helix B is broken, and cannot break the autoinhibition pocket. PCA analysis and porcupine plots also indicate that the EPIYA-D binding to the N-SH2 domain could cause SHP2 to have more motion modes, which could be conducive to breaking the autoinhibition pocket of SHP2. Our study explores the different abilities of EPIYA-D and EPIYA-C to activate SHP2. These results can help us better understand the mechanism of gastric cancer caused by *Helicobacter pylori* infection.

## Figures and Tables

**Figure 1 molecules-26-00837-f001:**
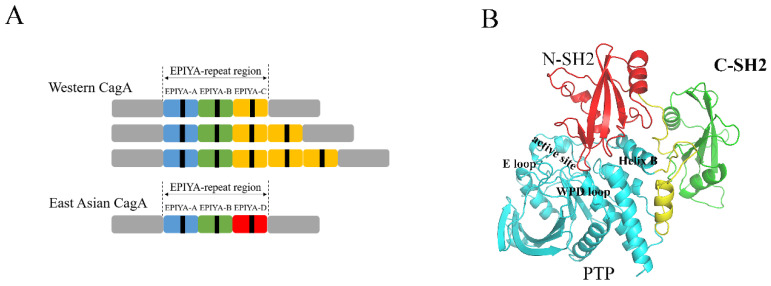
The structural diagram of *H. pylori* CagA (**A**). Each of the EPIYA segments contains a single EPIYA tyrosine phosphorylation motif (shown as a black box). The overall structure of SHP2 (**B**). The N-SH2 domain is shown in red, the C-SH2 domain is shown in green, and the PTP domain is shown in blue. The linker region is shown in yellow, and the active site and helix B are marked.

**Figure 2 molecules-26-00837-f002:**
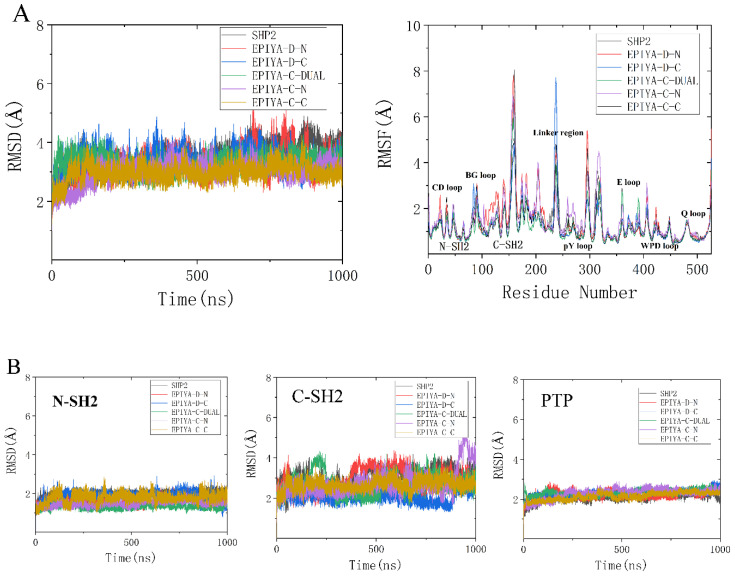
RMSD values and RMSF values of SHP2 in all systems (**A**), and the RMSD values of N-SH2, C-SH2 and PTP domains in all systems (**B**).

**Figure 3 molecules-26-00837-f003:**
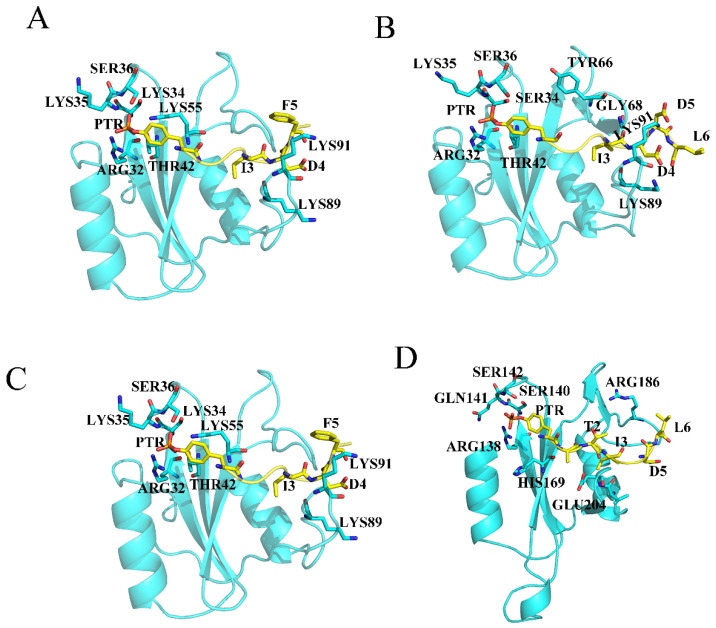
The binding details of systems EPIYA-D-N (**A**), EPIYA-C-N (**B**), EPIYA-D-N (**C**), and EPIYA-C-C (**D**). The SHP2 protein is shown in blue, and the peptides EPIYA are shown in yellow.

**Figure 4 molecules-26-00837-f004:**
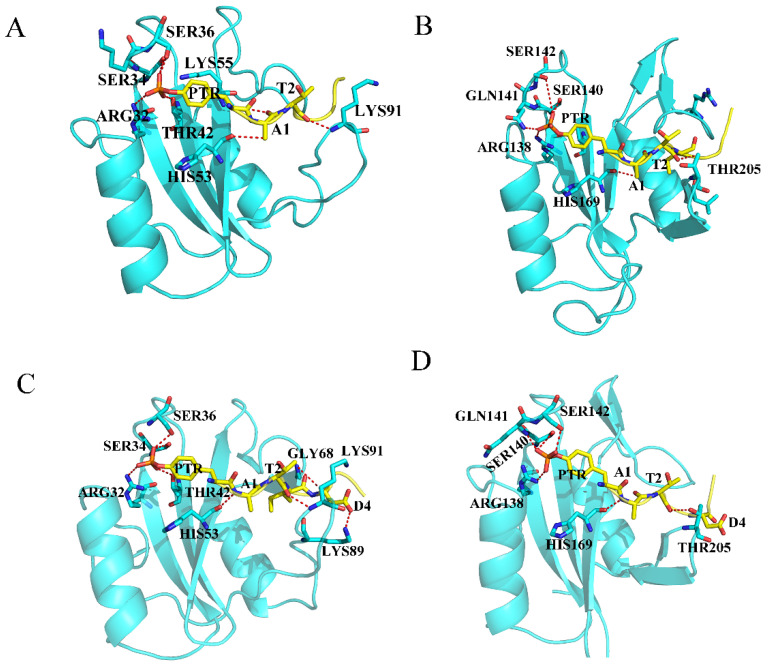
The hydrogen bonds between the SH2 domains and EPIYA in the systems EPIYA-D-N (**A**), EPIYA-D-C (**B**), EPIYA-C-N (**C**), and EPIYA-C-C (**D**). The residues on EPIYA peptides are identified by one letter (except PTR), and the residues on SHP2 are identified by three letters. The SHP2 protein is shown in blue, and the EPIYA peptides are re shown in yellow. The red dotted line means there is a hydrogen bond between the two residues.

**Figure 5 molecules-26-00837-f005:**
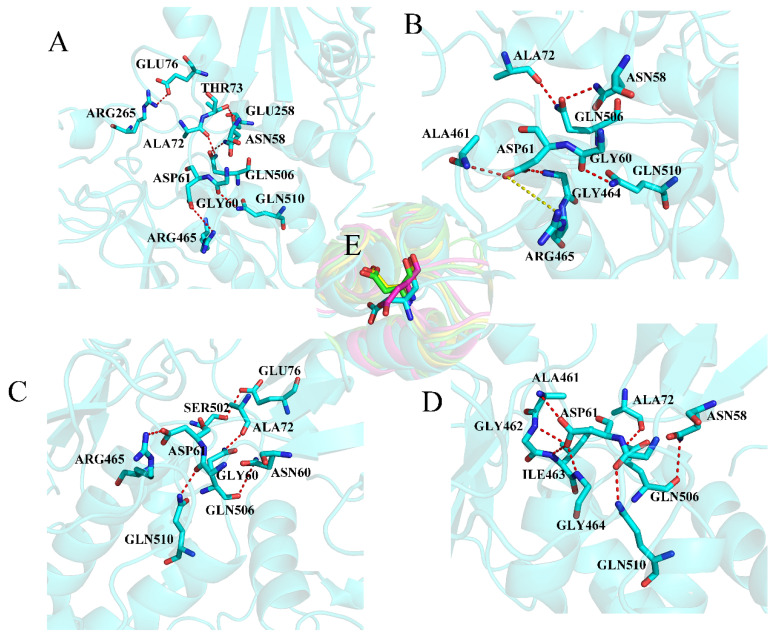
The hydrogen bonds between the N-SH2 domain and the PTP domain in the systems SHP2 (**A**), EPIYA-D-N (**B**), EPIYA-C-N (**C**), and EPIYA-C-DUAL (**D**). The red dotted line means that there is a hydrogen bond between the two residues, and the yellow dotted line means that the distance is too great to form a hydrogen bond (more than 3.5Å). The key residue ASP61 is shown separately for the systems SHP2 (magenta), EPIYA-D-N (green), EPIYA-C-N (blue), and EPIYA-C-DUAL (yellow) (**E**).

**Figure 6 molecules-26-00837-f006:**
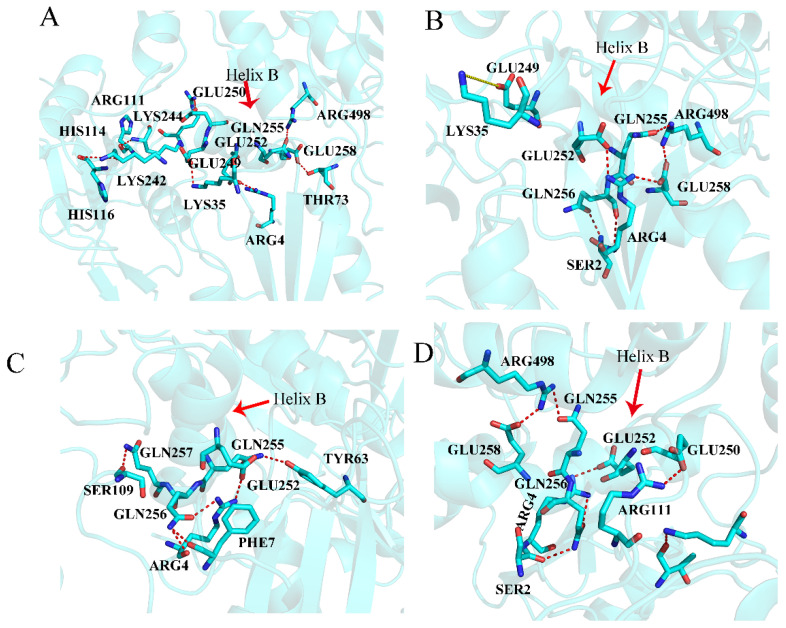
The hydrogen bonds of key helix B in system SHP2 (**A**), system EPIYA-D-N (**B**), system EPIYA-C-N (**C**), and system EPIYA-C-DUAL (**D**). The red dotted line means there is a hydrogen bond between the two residues, and the yellow dotted line means that the distance is too great to form a hydrogen bond (more than 3.5Å).

**Figure 7 molecules-26-00837-f007:**
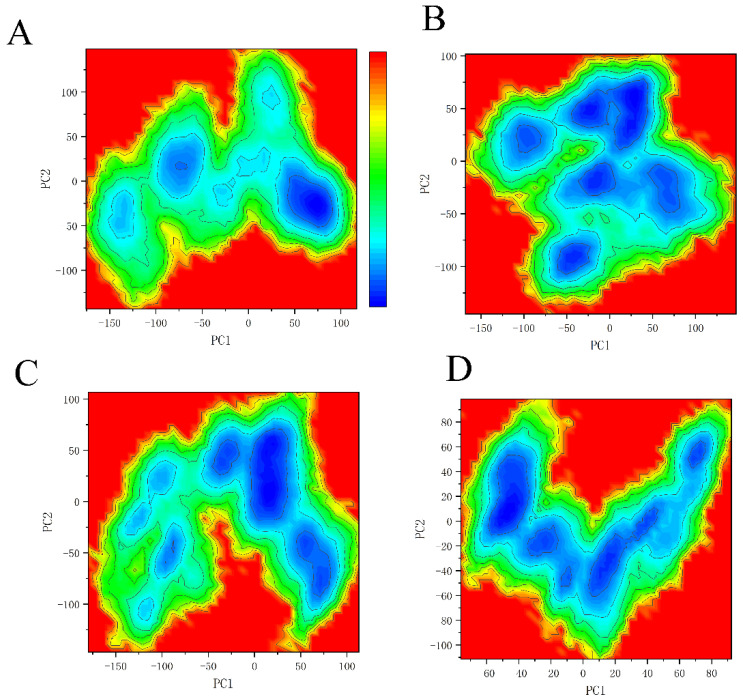
Free energy landscapes of system SHP2 (**A**), system EPIYA-D-N (**B**), system EPIYA-C-N (**C**), and system EPIYA-C-DUAL (**D**).

**Figure 8 molecules-26-00837-f008:**
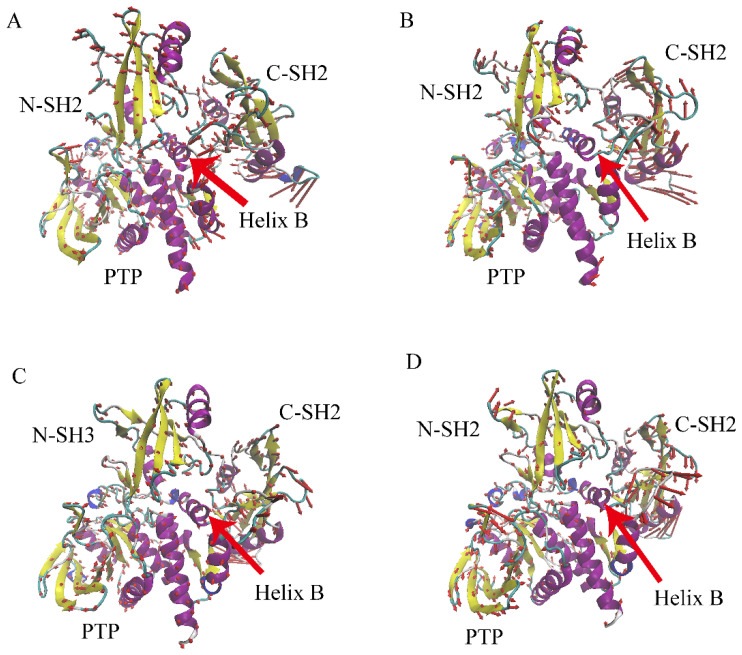
The porcupine plots of system SHP2 (**A**), system EPIYA-D-N (**B**), system EPIYA-C-N (**C**), and system EPIYA-C-DUAL (**D**).

**Figure 9 molecules-26-00837-f009:**
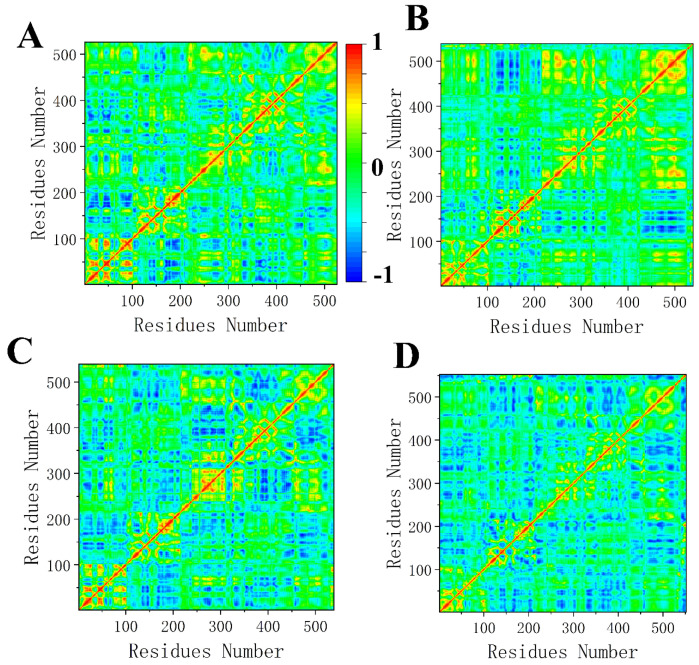
The cross-correlation maps of system SHP2 (**A**), system EPIYA-D-N (**B**), system EPIYA-C-N (**C**), and system EPIYA-C-DUAL (**D**). The covariance values are within the range of −1 to 1, where the 1 reflect the pairs of residues moving in the same direction. The −1 reflects those residues moving in the opposite direction.

**Table 1 molecules-26-00837-t001:** The RDOCK results of all systems. The results of the N-SH2 and C-SH2 domains for system EPIYA-DUAL are shown separately.

System	E_RDOCK (kcal mol^−1^)
EPIYA-D-N	−20.18
EPIYA-C-N	−14.42
EPIYA-D-C	−16.96
EPIYA-C-C	−14.79
EPIYA-DUAL-NSH2	−13.83
EPIYA-DUAL-CSH2	−13.42

**Table 2 molecules-26-00837-t002:** Binding free energies (kcal mol^−1^) and its components for complex systems.

	EPIYA-D-N	EPIYA-D-C	EPIYA-C-N	EPIYA-C-C	DUAL-NSH2	DUAL-CSH2
E_ele_	−733.97 ± 35.89	−369.97 ± 33.28	−791.76 ± 64.56	−312.49 ± 35.78	−488.49 ± 54.31	−216.28 ± 43.11
E_vdw_	−66.00 ± 5.45	−66.74 ± 5.15	−57.31 ± 6.68	−60.09 ± 5.22	−61.88 ± 5.25	−70.35 ± 5.21
G_gb_	713.94 ± 33.79	353.28 ± 29.13	773.24 ± 58.78	314.41 ± 32.73	473.16 ± 49.85	232.78 ± 40.21
G_SA_	−10.79 ± 0.53	−10.93 ± 0.31	−9.25 ± 0.72	−9.40 ± 0.54	−9.49 ± 0.49	−10.35 ± 0.36
^a^ G_pol_	−20.03 ± 49.29	−16.69 ± 44.23	−18.52 ± 87.31	1.92 ± 48.49	−15.33 ± 73.72	16.50 ± 58.95
^b^ G_nonpol_	−76.79 ± 5.48	−77.67 ± 5.16	−66.56 ± 6.72	−69.49 ± 5.25	−71.73 ± 27.80	−80.70 ± 5.22
H	−96.82 ± 8.72	−94.36 ± 9.45	−85.08 ± 13.03	−67.57 ± 8.81	−86.70 ± 11.68	−64.21 ± 6.17
TS	−53.55 ± 6.08	−55.67 ± 1.58	−68.02 ± 0.30	−47.70 ± 4.22	−48.55 ± 2.84	−47.05 ± 3.85
^c^ G_bind_	−43.27 ± 10.63	−38.69 ± 9.58	−17.06 ± 13.03	−19.87 ± 9.77	−38.15 ± 12.02	−17.16 ± 7.27

^a^ G_pol_ = E_ele_ + G_gb_. ^b^ G_nonpol_ = E_vdw_ + G_SA_. ^c^ G_bind_ = G_nonp_ + G_pol_ − TS.

## Data Availability

Data is contained within the article and Appendix A.

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
