# Peer review of "Exploring the Distinct Binding and Activation Mechanisms for Different CagA Oncoproteins and SHP2 by Molecular Dynamics Simulations"

_molecules, 2021, doi:10.3390/molecules26040837_

Round 1
Reviewer 1 Report
Wang et al. have investigated through molecular dynamics (MD) simulations the recognition of different CagA oncoprotein segments by SHP2. Previous experimental work from Hayashi et al. reported on different affinities of EPIYA segments towards the pro-oncogenic SHP2 phosphatase. In the current manuscript, the authors give further insights into the molecular mechanism underlying different EPIYA segment effects on SHP2 activity through a comprehensive analysis of their MD data. The manuscript is well organized, the data are well presented and the references are appropriate. Some additional English editing is recommended. Overall, this reviewer finds enough merit to publish this work in the Molecules journal and will recommend it for publication after minor revision.
Starting with the Figures, their quality should be highly improved in terms of resolution. In addition, I include below some suggestions for increasing the clarity of some of them:
Fig 1B. Text inside of SHP2 protein structure referring its most relevant regions such as “helix B”, active site, etc, could be more easily readable if they would appear in bold. The same for Figures 3-6. When possible, increase the size of the font used.
Fig 2 fully lacks of good resolution. For this reviewer is not possible to properly seeing the color line codes related to each simulated system legend. The width of each line should be increased. In addition, the numbers in the x-y axes and their corresponding legends are too small. In Fig. 2A, the last number appears as 100, and it should be corrected to 1000.
Fig. 5 legend has two typos in lines 198/199 and 201. Lines 198/199: “The red dotted line means there is” should be corrected to “The red dotted line means that there is”. Line 201: “bule” should be corrected to “blue”.
With respect to Figures 5 and 6, which criteria was used by the authors to define H-bond formation in their MD simulations? This information should appear in the Methods section. The authors should define more precisely in the manuscript/figure legends when a distance between two atoms is too long to form a H-bond.
Fig. 9: same comment as in Fig 2 with respect to the numbers in the x-y axes and their corresponding legends. In addition, the heat-column in Panel A at the right, which includes color-codes used in the cross-correlation maps, lacks the corresponding numbers associated to such color codes. This should be introduced accordingly.
Table 1 has several typos. The superscripts “a” referring to Gpol, “b” referring Gnonpol and “c” referring Gbind should be properly introduced in the first column of the table. In addition, in the equation cGbind = Gnonp + Gpb – TS, Gnonp and Gpb should be corrected to Gnonpol and Gpol, respectively.
When referring to “conformation change” the authors should correct it to “conformational change” (it appears several times along the manuscript).
Which software was used to perform the analysis of the MD data? This information should be also included in the Methods section.
Author Response
Thanks for reviewers’ valuable comments and for pointing out the problems in this manuscript. We have corrected these problems and have improved inadequacies as requested. And we have responded to reviewers’ comments in this file in detail.
Response to Reviewer 1 Comments
Reviewer1
Wang et al. have investigated through molecular dynamics (MD) simulations the recognition of different CagA oncoprotein segments by SHP2. Previous experimental work from Hayashi et al. reported on different affinities of EPIYA segments towards the pro-oncogenic SHP2 phosphatase. In the current manuscript, the authors give further insights into the molecular mechanism underlying different EPIYA segment effects on SHP2 activity through a comprehensive analysis of their MD data. The manuscript is well organized, the data are well presented and the references are appropriate. Some additional English editing is recommended. Overall, this reviewer finds enough merit to publish this work in the Molecules journal and will recommend it for publication after minor revision.
Fig 1B. Text inside of SHP2 protein structure referring its most relevant regions such as “helix B”, active site, etc, could be more easily readable if they would appear in bold. The same for Figures 3-6. When possible, increase the size of the font used.
Response 1: Thanks for reviewer’s valuable suggestions. We have improved these figures as requested. These texts in figures were set in bold, and we increased these font size appropriately.
Fig 2 fully lacks of good resolution. For this reviewer is not possible to properly seeing the color line codes related to each simulated system legend. The width of each line should be increased. In addition, the numbers in the x-y axes and their corresponding legends are too small. In Fig. 2A, the last number appears as 100, and it should be corrected to 1000.
Response 2: We have improved these figures as requested. The resolution of Fig3 was increased. We increase the width of each line, and increase the font size of the number of x-y axes. The last number “100” was corrected to “1000”
Fig. 5 legend has two typos in lines 198/199 and 201. Lines 198/199: “The red dotted line means there is” should be corrected to “The red dotted line means that there is”. Line 201: “bule” should be corrected to “blue”.
Response 3: Thanks for reviewer’s proofreading. We have corrected these typos according to suggestions.
With respect to Figures 5 and 6, which criteria was used by the authors to define H-bond formation in their MD simulations? This information should appear in the Methods section. The authors should define more precisely in the manuscript/figure legends when a distance between two atoms is too long to form a H-bond.
Response 4: We have added the criteria which is used by us to define H-bond formation in the paragraph “3.3. Analysis of Hydrogen Bonds”. We have defined a distance more than 3.5Å which is too long to form an H-bond between two atoms. And we have added these descriptions in the legends of Figures 5 and 6.
Analysis of Hydrogen Bonds
Hydrogen bonds in all systems were analyzed using the cpptraj module of AMBER16. The presence of these bonds was calculated over the last 300ns of the simulations. The criteria used for hydrogen bonding was that the distance was less than 3.5Å between the hydrogen bond acceptor and the hydrogen bond donor, and the angle of acceptor-H-donor was more than 120°.
Fig. 9: same comment as in Fig 2 with respect to the numbers in the x-y axes and their corresponding legends. In addition, the heat-column in Panel A at the right, which includes color-codes used in the cross-correlation maps, lacks the corresponding numbers associated to such color codes. This should be introduced accordingly.
Response 5: We have improved these figures as requested. We increase the font size of the number and legend of x-y axes. And the corresponding numbers were added to the Panel A. In addition, we added an introduction in the figure legend.
The covariance values are within the range of -1 to 1, where the 1 reflect the pairs of residues moving in the same direction. And the -1 reflect those residues moving in opposite direction.
Table 1 has several typos. The superscripts “a” referring to Gpol, “b” referring Gnonpol and “c” referring Gbind should be properly introduced in the first column of the table. In addition, in the equation cGbind = Gnonp + Gpb – TS, Gnonp and Gpb should be corrected to Gnonpol and Gpol, respectively.
Response 6: The former Table 1 has been changed to Table 2. We have introduced the superscripts “a”, “b” and “c” in the first column of the table. And we have corrected the typos in the equation and in Table 2.
When referring to “conformation change” the authors should correct it to “conformational change” (it appears several times along the manuscript).
Response 7: We have corrected all these errors.
Which software was used to perform the analysis of the MD data? This information should be also included in the Methods section.
Response 8: We have added this information in the paragraph “3.2. Molecular Dynamics Simulations” (from line 332 to line 335).
Most of the MD data analysis were performed by the cpptraj module of AMBER 16. The binding free energy was calculated using the molecular mechanics Generalized Born Surface Area (MM-GB/SA ) method by AMBER 16. The porcupine plots analysis was performed by the ProDy interface in the VMD software.

Reviewer 2 Report
Wang et al constructed a model of the interaction between SHP2 and EPIYA segments present on CagA, which is considered a major virulence factor of Helicobacter pylori. Through MD simulation the activation mechanism is investigated with a deep analysis of the allosteric regulation. The work is well conducted, and I raise only few things.
- Starting structures affect molecular dynamics (MD) simulations results. For this reason molecular docking results should be reported in the manuscript and the authors show how the starting structures for MD were generated.
- Paragraph 3 is a sort of conclusion section rather than a discussion one. Reorganization of this part of the manuscript is welcome.
- Quality and resolution of Figure 2 should be improved.
- Line 112, correct the sentence
- Line 307 replace Sdudio with Studio
Author Response
- Starting structures affect molecular dynamics (MD) simulations results. For this reason molecular docking results should be reported in the manuscript and the authors show how the starting structures for MD were generated.
Response 1: Thanks for reviewer’s valuable suggestions. We have shown how the starting structures were generated and have added the molecular docking results in the manuscript in a new paragraph “2.1. The Molecular Docking Analysis” and new Table 1.
2.1. The Molecular Docking Analysis
First, ZDOCK was used to obtain the initial docking poses, and then RDOCK was used to optimize and choose the final docking poses (Table 1). The structure inspection revealed there is no unreasonable structure. Compared with the crystal structures (PDB codes: 5X7B, 5X94), the RMSD values of the RDOCK results are all less than 1.5Å. These results indicate that the docking results are credible, and can be used as the initial structure for subsequent MD simulations.
Table 1. The RDOCK results of all systems. The results of N-SH2 and C-SH2 domains of system EPIYA-DUAL were shown separately.
|
System |
E_RDOCK (kcal mol-1) |
|
EPIYA-D-N |
-20.18 |
|
EPIYA-C-N |
-14.42 |
|
EPIYA-D-C |
-16.96 |
|
EPIYA-C-C |
-14.79 |
|
EPIYA-DUAL-NSH2 |
-13.83 |
|
EPIYA-DUAL-CSH2 |
-13.42 |
- Paragraph 3 is a sort of conclusion section rather than a discussion one. Reorganization of this part of the manuscript is welcome.
Response 2: We have reorganized this part as a conclusion and the discussion have been integrated into the paragraph “2. Results and discussion”. We suppose that this adjustment is going to make this manuscript be more in line with the convention of publication in this journal.
- Quality and resolution of Figure 2 should be improved.
Response 3: We have improved this figure as requested.
- Line 112, correct the sentence.
Response 4: Thanks for reviewer’s proofreading. We have corrected this sentence in line 123 now (“and some loops in PTP domain also show a high level of flexibility.”).
5 Line 307 replace Sdudio with Studio.
Response 5: We have corrected this mistake.
Besides, we have also corrected some typos and errors which are found by ourselves. The words highlighted in yellow indicate the trace where we modified. These corrections are shown as follow:
- Line 113 (which indicates there are no obvious structure changes having occurred under the simulations in all systems.).
- Line 127 (the flexibility of E loop and pY loop is weaker than that of system EPIYA-C-N.).
- Line 150. We have corrected the caption of Figure 3 (B) and (C).
- Line 203 (ASP61 does not flip).
- Line 246 (EPIYA-D binding to N-SH2 domain of SHP2).
- Line 266 (but the motion modes of C-SH2 and PTP domains are the same with that of system SHP2.).
- Line 280 (and it represents these pairs of residues).
- Line 286 (System EPIYA-C-DUAL (D) has a similar correlation map).
- Line 306 (the C-SH2 domain of SHP2).
- Line 307 (C-SH2 domain of SHP2).
- Line 330 (All bonds involving hydrogen atoms were held fixedly).
- Line 346 (the stabilization energy due to solvation,).
- Line 352 (0.0072 kcal mol-1 Å-2 and 0.00 kcal mol-1.).
- Line 373 (In FEL, free energy minima represents stable conformations).
- We have corrected a citation error of reference 22.

Round 2
Reviewer 2 Report
The authors have addressed my comments